# “Pleiotropic” Effects of Antibiotics: New Modulators in Human Diseases

**DOI:** 10.3390/antibiotics13121176

**Published:** 2024-12-04

**Authors:** Carlo Airola, Andrea Severino, Irene Spinelli, Antonio Gasbarrini, Giovanni Cammarota, Gianluca Ianiro, Francesca Romana Ponziani

**Affiliations:** 1Digestive Disease Center, Fondazione Policlinico Universitario Agostino Gemelli IRCCS, 00168 Rome, Italy; carlo.airola01@icatt.it (C.A.); andrea.severino01@icatt.it (A.S.); irene.spinelli@guest.policlinicogemelli.it (I.S.); antonio.gasbarrini@unicatt.it (A.G.); giovanni.cammarota@unicatt.it (G.C.); gianluca.ianiro@unicatt.it (G.I.); 2Dipartimento Universitario di Medicina e Chirurgia Traslazionale, Università Cattolica del Sacro Cuore, Largo Francesco Vito 1, 00168 Rome, Italy

**Keywords:** eubiotics, immunomodulators, microbiota, immune system

## Abstract

Antibiotics, widely used medications that have significantly increased life expectancy, possess a broad range of effects beyond their primary antibacterial activity. While some are recognized as adverse events, others have demonstrated unexpected benefits. These adjunctive effects, which have been defined as “pleiotropic” in the case of other pharmacological classes, include immunomodulatory properties and the modulation of the microbiota. Specifically, macrolides, tetracyclines, and fluoroquinolones have been shown to modulate the immune system in both acute and chronic conditions, including autoimmune disorders (e.g., rheumatoid arthritis, spondyloarthritis) and chronic inflammatory pulmonary diseases (e.g., asthma, chronic obstructive pulmonary disease). Azithromycin, in particular, is recommended for the long-term treatment of chronic inflammatory pulmonary diseases due to its well-established immunomodulatory effects. Furthermore, antibiotics influence the human microbiota. Rifaximin, for example, exerts a eubiotic effect that enhances the balance between the gut microbiota and the host immune cells and epithelial cells. These pleiotropic effects offer new therapeutic opportunities by interacting with human cells, signaling molecules, and bacteria involved in non-infectious diseases like spondyloarthritis and inflammatory bowel diseases. The aim of this review is to explore the pleiotropic potential of antibiotics, from molecular and cellular evidence to their clinical application, in order to optimize their use. Understanding these effects is essential to ensure careful use, particularly in consideration of the threat of antimicrobial resistance.

## 1. Introduction

The introduction of antibiotics in the 1930s revolutionized the field of medicine by providing a highly effective treatment for infectious diseases, leading to a dramatic improvement in human life expectancy [1]. During the “golden era” of antibiotic research, which lasted from the 1950s to the 1970s, numerous new classes of antibiotics were discovered. Between 2000 and 2018, the global antibiotic consumption rate increased from 9.8 defined daily doses (DDD) per 1000 people per day to 14.1 DDD per 1000 people per day [2]. Since then, the focus of research has switched to improving the efficacy of existing antibiotics [3]. The success of antibiotics can be attributed to their principal mechanism of action, which involves directly inhibiting or killing microorganisms. Some antibiotics, however, carry auxiliary features contributing to their extensive clinical usage [4]. These additional positive benefits, which also occur for other drugs’ classes, such as statins, are known as pleiotropic effects [4]. For instance, certain antibiotics, such as clindamycin and linezolid, have antitoxin effects in addition to antibacterial properties [5,6]. Nonetheless, some antibiotics exert a direct impact on the immune system, which could be crucial in the host’s response to an infection. Among the different classes of antibiotics, those that affect protein synthesis or DNA regulation, like macrolides, tetracyclines, and fluoroquinolones, seem to be more implicated in a pleiotropic effect on human immune cells, while antibiotics inhibiting bacterial wall synthesis seem to be less involved in immunomodulation [7,8,9]. Nonetheless, antibiotics interfere with the billions of bacteria that normally inhabit the human body, in addition to their direct immunomodulatory action. In most situations, antibiotics disrupt human microbiota; nevertheless, certain antimicrobial drugs have been discovered to improve microbiota composition and restore balance in dysbiotic circumstances, which can favorably affect systemic clinical disorders [10].

The plethora of adjunctive effects of antibiotics unveils new therapeutic possibilities for this class of drugs. For the role they could have in the treatment of several human diseases, this review aims to investigate the primary pleiotropic effects of antibiotics, with a particular emphasis on their immunomodulatory actions, involving molecular processes and cell responses, and their influence on microbiota modification. The dual impact on the immune system and the human microbiome might open new routes for treating human illnesses. It is critical to thoroughly assess the molecular processes implicated and examine the therapeutic consequences, especially in the setting of antimicrobial resistance (AMR). Such comprehension will aid in identifying specific patients who will benefit from these medicinal applications.

## 2. Immunomodulatory Effect of Antibiotics in Critical Illnesses

Antibiotics, due to their antimicrobial activity, have been used to treat bacterial infections in patients since their discovery [11]. However, it is crucial to remember that the host immune system has a substantial impact on the outcome of infections and their consequent death rates [12]. A dysregulated immune response in the host can result in systemic complications, such as sepsis, the most lethal infection consequence [13].

Early antibiotic therapy has been shown to significantly reduce sepsis-related mortality [14], but despite the effective eradication of the causal infection, many patients with sepsis still die [13]. The host’s systemic inflammatory response to infection appears to be the key contributing factor [13]. Pathogens activate immune cells during infection by interacting with pattern-recognition receptors (PRRs), which are part of complexes known as inflammasomes [15]. These receptors identify pathogen-associated molecular patterns (PAMPs), which are shared by many microbial species, causing inflammatory gene transcription to increase and the innate immune response to start. Endogenous molecules called damage-associated molecular patterns (DAMPs), which are generated from injured cells during infection, can also trigger this molecular pathway [16]. The activation of PRRs induces immune cells to combat invading pathogens. However, critically ill patients often experience an excessive inflammatory response marked by an overwhelming release of pro-inflammatory cytokines. This uncontrolled inflammation significantly contributes to the development of sepsis. Interestingly, preclinical models of sepsis have demonstrated that antibiotics can enhance survival, even when sepsis is induced by exposure to lipopolysaccharide (LPS) or multidrug-resistant pathogens. In such cases, the effectiveness of antibiotics should not be solely attributed to their antimicrobial activity. Instead, a growing body of evidence suggests a direct impact of antibiotics on the immune system, such as in the case of macrolides, tetracyclines, and fluoroquinolones [17,18,19]. Therefore, while antibiotics are renowned for their potent antibacterial properties, they may also possess a direct immunomodulatory effect, contributing to their remarkable efficacy.

## 3. Antibiotic Modulation of Chronic Inflammation

The immunomodulatory effects of specific classes of antibiotics extend beyond acute infections and have been increasingly recognized in chronic conditions, particularly respiratory diseases. Notably, macrolides have historically played a role in asthma to reduce reliance on corticosteroids [20]. They have also been beneficial in other inflammatory respiratory diseases such as panbronchiolitis, where they improve long-term survival [21]. Interestingly, in these chronic conditions, the therapeutic effects of antibiotics appear to be associated with their non-antimicrobial properties [22].

In April 2020, the British Thoracic Society issued guidelines recommending the use of long-term, low-dose macrolide antibiotics for adult respiratory diseases, highlighting their role in immunomodulation. These guidelines provide specific recommendations for conditions such as adult asthma, bronchiectasis, chronic obstructive pulmonary disease (COPD), bronchiolitis obliterans (BO), and other respiratory diseases [23]. At the molecular level, chronic pulmonary diseases, as well as chronic systemic inflammation, involve the release of damage-associated molecular patterns (DAMPs) due to tissue damage and the activation of Toll-like receptor (TLR) pathways [24,25,26]. Furthermore, conditions like asthma and COPD seem to involve the NOD-like receptor protein 3 (NLRP3) inflammasome, activated by DAMPs and pathogen-associated molecular patterns (PAMPs), leading to an increased production of pro-inflammatory cytokines by various immune cells [27]. DAMPs also trigger M1 pro-inflammatory macrophages, resulting in an increased production of tumor necrosis factor alpha (TNF-α) [24]. This underlying process in chronic pulmonary diseases serves as a pathophysiological model applicable to various other chronic illnesses, including chronic pain [28], Alzheimer’s and Parkinson’s disease [29,30], coronary artery disease [31], insulin resistance [32], non-alcoholic steatohepatitis [33], liver fibrosis progression, inflammatory bowel diseases (IBDs) [34], and rheumatoid arthritis [35]. Interestingly, certain antibiotics may modulate these pathways and molecules, providing potential therapeutic benefits. Moreover, matrix metalloproteinases (MMPs), involved in cleaving extracellular proteins, appear to be consistently activated in various chronic inflammatory conditions, particularly in COPD [36]. MMPs, induced by Toll-like receptor 5 (TLR5) signaling, also seem to play a crucial role in the development of IBDs [37]. Additionally, in colitis models, MMPs contribute to bridging innate and adaptive immunity by regulating CD8+ T cells, which play a significant role in IBD pathogenesis [38]. MMPs are overexpressed in conditions like rheumatoid arthritis, along with interleukin 1β (IL-1β), interleukin 6 (IL-6), and chemokines [39]. Remarkably, certain antibiotics have the potential to modulate these molecules as well. Overall, the immunomodulatory effects of antibiotics extend into the realm of chronic diseases, paving the way for their therapeutic application in conditions marked by chronic inflammation and immune dysregulation (Figure 1).

## 4. Macrolides as Versatile Immunomodulators: Insights into Their Effects on Acute and Chronic Inflammation

Macrolides are recognized for their immunomodulatory action, particularly in the case of chronic lung diseases [7,8]. Nonetheless, macrolides have also demonstrated promising results as adjuvants to reduce acute systemic inflammation. Studies have shown that clarithromycin, azithromycin, and telithromycin effectively suppress inflammation and increase survival in experimental sepsis induced by LPSs [40,41,42]. Moreover, in a double-blind trial involving 200 patients with sepsis and ventilator-associated pneumonia, the intravenous administration of clarithromycin for 3 days resulted in an earlier resolution of pneumonia, a reduced duration of mechanical ventilation, and delayed progression to multiple organ dysfunction syndrome (MODS). However, there was no significant difference in 28-day mortality with the placebo group [43]. On the contrary, a subsequent analysis showed that mortality between 28 and 90 days was significantly reduced in the clarithromycin group [44]. In another trial involving 600 individuals with suspected or microbiologically confirmed Gram-negative sepsis, administration of clarithromycin led to a reduction in symptom duration, particularly in critically ill patients [45]. It is important to note that both clinical studies specifically included patients whose causative bacteria were resistant to macrolides [43,45]. Conversely, a recent clinical trial comparing clarithromycin to a placebo in sepsis patients revealed a higher risk of acute kidney injury in the clarithromycin group, although there were no significant differences in 28-day mortality [46]; however, the results of this trial were limited by its small sample size. Indeed, while clarithromycin is not typically associated with renal toxicity, it is a well-known cytochrome P450 inhibitor [47] and may interact with the metabolism of nephrotoxic drugs, increasing the risk of renal injury. Unfortunately, the specific drugs co-administered with clarithromycin were not reported in the study. Nevertheless, it appears that this observed higher risk of acute kidney injury may not be a drug class effect. In fact, a retrospective study demonstrated that early exposure to azithromycin was associated with a lower incidence of major adverse kidney events in patients with sepsis-associated acute kidney injury [48]. Unlike other macrolides, azithromycin does not interfere with cytochrome metabolism [49]. Macrolides seem to dampen systemic inflammation in some critical conditions, as shown by clinical evidence. In fact, according to serum analysis performed by Spyridaki et al., patients with septic shock and multiple organ failure who received clarithromycin had a lower ratio of serum IL-10 to TNF-α, although their monocytes showed higher IL-6 but lower TNF-α production in response to LPSs with increased apoptosis, and increased expression of the co-stimulatory molecule CD86 on day 4 after randomization [50]. These results may point out that clarithromycin could prevent a state of immunoparalysis in the early phases of systemic inflammation with endotoxin tolerance, leading to homeostasis in the late phases. This agrees with the results obtained in patients with chronic lung diseases, in whom both azithromycin and clarithromycin administration increased serum IL-8 levels during the first days of treatment, with a subsequent decrease. In the case of clarithromycin, granulocyte–macrophage colony-stimulating factors (GM-CSFs) also followed this kinetic [51].

At a molecular level, the macrolides’ immunomodulatory effect seems to be particularly related to TLRs, in which recognizing PAMPs and DAMPs are critical to enhance systemic inflammation. For instance, azithromycin reduces TLRs’ surface expression on dendritic cells [52,53] and macrophages [54], inducing an anti-inflammatory phenotype in these cells. Nonetheless, preclinical studies confirmed what serum analysis demonstrated in macrolide-treated patients. Macrolides reduce the production of pro-inflammatory cytokines, such as IL-6, IL-12, and TNF-α, in airway epithelial cells, monocytes, and dendritic cells [53,55,56]. Analogous results were obtained in bacterial infection preclinical models, in which macrolide administration was associated with a reduction in IL-1β, IL-6, and TNF-α levels in serum and bronchoalveolar lavage fluid (BALF) [57,58,59]. Nonetheless, macrolides inhibit NLRP3 and NLR family CARD domain containing 4 (NLRC4) inflammasome activation, which diminishes the production of IL-1β by monocytes and macrophages in response to either lipopolysaccharide stimulation or whole bacteria [60,61]. In a recent study, the effect of in vitro neutrophils loaded with azithromycin and colistin was tested in *Pseudomonas aeruginosa* infection in mice. Interestingly, pre-treated neutrophils conserved antimicrobial activity, migration ability, and reactive oxygen species production, but downregulated pro-inflammatory cytokine production, such as IL-1β, IL-6, and TNF-α [62].

While macrolides’ capacity to inhibit pro-inflammatory cytokine production seems well established, their effect on anti-inflammatory cytokines may be ambivalent. In fact, macrolides upregulate IL-10 production by monocytes and macrophages, inducing the shift into an M2 anti-inflammatory phenotype of the latter [55,63], but seem to suppress the production of IL-10 in T cells and dendritic cells [52]. A recent trial, in which azithromycin administration attenuated post-inflammatory wheezing in children, showed that its clinical benefit was associated with the inhibition of the hypermethylation of histone H3K27me3 mediated by the enhancer of zest homolog 2 (EZH2), increasing IL-10 serum levels [64]. In another recent preclinical trial, azithromycin mitigated cisplatin-induced lung damage in mice by increasing IL-10 blood levels through the stimulation of Nrf2/HO-1, cytoglobin, peroxisome proliferator-activated receptor γ (PPARγ), and sirtuin 1 (SIRT1) pathways [65]. Macrolides also reduce immune cell recruitment. Azithromycin and clarithromycin administration in mouse models of acute infection and high-pressure ventilation lung injury resulted in a reduced recruitment of leukocytes, particularly neutrophils, to the lung, and consequently prevented tissue destruction [66,67]. As a consequence, macrolides could also dampen chronic inflammation, in which neutrophils could be predominantly involved [68]. It is noteworthy that macrolides suppress MMP-9 production by neutrophils and other cells, such as fibroblasts, in inflammatory preclinical models [69,70]. Indeed, MMP-9, together with other MMPs, promotes the migration of neutrophils, dendritic cells (DCs), and other leukocytes by interacting with chemokines and producing isoforms with increased biological activity [71]. Among others, chemokines like the chemokine (C-X-C motif) ligand 5 (CXCL5) and CXCL8 seem to be particularly involved in this process [71]. In fact, azithromycin, clarithromycin, and roxithromycin downregulate chemokines, such as CXCL5 and CXCL8, which are crucial in neutrophil migration [72,73], in cells isolated from the sputum samples of patients with COPD [74]. Furthermore, macrolide treatment reduces neutrophil survival by directly stimulating apoptosis and by inhibiting the release of pro-survival molecules such as the granulocyte–macrophage colony-stimulating factor (GM-CSF) [75,76]. However, azithromycin seems to preserve neutrophil lifespan in the presence of bacteria, such as *Streptococcus pneumoniae*.

Macrolide treatment also interferes with other immune cells’ behavior. Indeed, azithromycin promotes macrophages to switch to the anti-inflammatory M2 phenotype in vitro and in vivo [77,78,79]. In addition, macrolides downregulate the expression of co-stimulatory molecules on antigen-presenting cells [52,80,81] and azithromycin (but not clarithromycin) reduces the expression of major histocompatibility complex class II on dendritic cells (DCs) in vitro [53]. Thus, macrolides could inhibit DC-dependent CD4+ T cell activation [52,82]. In addition, CD4+ T cells exposed to DCs pre-treated with azithromycin produce less interferon γ (IFN-γ) and more IL-10 [53]. It is noteworthy that macrolides, in particular azithromycin, directly suppress CD4+ T cells in vitro by modulating the mTOR pathway [83]. Nonetheless, rapamycin, a macrolide that targets mTOR, was approved in 1999 as an immunosuppressant drug for the prevention of organ rejection in kidney transplant patients [84].

A recent multi-site double-blind, placebo controlled, randomized controlled trial showed that a 48-week weekly azithromycin treatment significantly reduced C reactive protein (CRP), MMP10, and neutrophil adhesion molecule E-selectin serum levels in individuals aged 6–19 years with HIV-associated chronic lung disease [85]. However, this effect disappeared 24 weeks after the cessation of the treatment. The OPTIMIZE randomized controlled trial confirm these data in children with cystic fibrosis and *P. aeruginosa* airway colonization, showing that azithromycin therapy significantly reduced CRP serum levels after 39 weeks compared to placebo. Even in this case, the cessation of the therapy was associated with the increase in CRP to the pre-treatment levels [86].

Finally, macrolides exert a well-documented immunomodulatory activity, which has been observed both in acute and chronic inflammation. Since most of the evidence derives from lung diseases, they gained approval in the treatment of chronic pulmonary diseases. Nonetheless, based on molecular and cellular evidence, further clinical trials could extend the use of macrolides to the treatment of other acute and chronic inflammatory conditions.

## 5. Tetracycline Impact on Inflammation Targeting the Inflammasome and MMPs

Starting from the paradigmatic experience of macrolides, other antibiotics have been evaluated for their immunomodulatory properties. Even tetracyclines showed an immunomodulatory effect and have been evaluated in some critical conditions, such as the acute respiratory distress syndrome (ARDS).

Peukert et al. described the effect of tetracycline on the NLRP3 inflammasome pathway in patients with ARDS. Human alveolar leukocytes were isolated within 24 h of onset of direct ARDS. Cultured leukocytes continued to produce IL-1ß and IL-18, suggesting that the NLRP3 inflammasome pathway remained intact. Tetracycline inhibited the production of IL-1ß and IL-18 by alveolar leukocytes in a dose-dependent manner through the inhibition of caspase-1 [87]. Moreover, in a preclinical trial, doxycycline reduced chemical acute lung injury by reducing neutrophil infiltration [88] and prevented neutrophil infiltration in models of virus-, bacteria-, and pancreatitis-induced ARDS [89,90,91]. In fact, doxycycline reduced the LPS-induced gene expression of IL-1ß and IL-6, as well as the phagocytosis of heat-inactivated *Escherichia coli* by monocytes [92]. Minocycline significantly ameliorated the LPS-induced inflammatory response in human monocytes obtained from healthy volunteers by decreasing the production of TNF-α, IL-1β, IL-6, cyclooxygenase 2 (COX-2), and prostaglandin E2 (PGE2). The immunomodulatory effects of minocycline were mediated through the inhibition of the nuclear factor kappa-light-chain-enhancer of activated B cell (NF-κB), the mitogen-activated protein kinase (MAPK), and the phosphoinositide 3-kinase (PI3K)/Akt pathways [93]. Furthermore, chemically modified tetracycline 3 (COL-3), which does not have an antimicrobial effect, decreased neutrophil infiltration and myeloperoxidase levels when administered to LPS-induced ARDS rat models [94]. Finally, nano-chemically modified tetracycline-3 (nCMT-3), when administered to mice before an LPS-induced acute lung injury (ALI), reduced histologic lung injury (polymorphonuclear neutrophil infiltration, alveolar thickening, edema, and consolidation), MMP-2 and -9 levels and activity, and NLRP-3 protein and activated caspase-1 levels in lung tissue. Moreover, nCMT-3 reduced plasma and BALF levels of the soluble triggering receptor expressed on myeloid cell 1 (sTREM-1), TNF-α, IL-1β, IL-6, IL-18, polymorphonuclear neutrophil (PMN) elastase, and BALF protein, decreasing the incidence of ALI [95].

A similar molecular mechanism seems to be involved in tetracycline’s effect on subacute and chronic inflammatory diseases. In fact, NLP3 inflammasomes and the consequent activation of the IL-1β and IL-18 pathway seem to also be crucially involved in chronic inflammatory lung diseases [96,97,98]. When exposed to silica particles in vitro, macrophages treated with tetracycline displayed reduced IL-1β production. Furthermore, these cells showed decreased pyroptotic cell death through the inhibition of caspase-1 compared to non-treated counterparts [99]. Moreover, a treatment of silica-instilled mice with tetracycline significantly reduced pulmonary caspase-1 activation as well as IL-1β and IL-18 production, thereby ameliorating pulmonary inflammation and lung injury [99]. In addition to the activation of inflammasomes, silicosis is notable for its elevation in MMP production, which can be observed in serum, particularly during the early stages of inflammation [100,101]. Tetracycline has been demonstrated to impact this component of chronic inflammation as well. In fact, the FDA in the United States has authorized doxycycline as an MMP inhibitor for the treatment of periodontal disease [102]. Tetracyclines have shown promising results in other clinical settings in which MMPs are implicated. Long-term therapy with doxycycline has been associated with a reduction in MMP8 in serum and body fluids containing inflammatory exudates in patients with reactive arthritis [103]. Minocycline is known for its MMP-inhibitory effect in neurological diseases [102]; a clinical trial showed that minocycline administration decreased plasma MMP-9 levels in patients with ischemic stroke [104]. Higher levels of MMP-9 in patients with ischemic stroke are associated with severe brain edema and hemorrhagic transformation [105] and predict bleeding after tissue plasminogen activator (t-PA) treatment [106]. Minocycline seems to exert a neuroprotective effect in case of ischemic stroke [107], but also other inflammatory neurological diseases, such as multiple sclerosis and autoimmune encephalomyelitis [108,109]. Interestingly, minocycline seems to reduce the ischemic stroke extent in preclinical models by decreasing the TNFα content in the penumbra area [110]. In fact, polymorphonucleocytes, which are highly implicated in ischemic brain injury [111], produce more MMP-9 with a neurotoxic effect in the presence of higher levels of TNFα [112]. In conclusion, tetracyclines have an immunomodulatory activity both in acute and chronic conditions by acting directly on key inflammatory factors, such as NLRP3 inflammasome, or by dampening MMP activity, which is widely implicated in immune response regulation, by cytokine proteolytic activation, vascular permeability modulation, and macrophage phagocytosis [113]. However, considering the complexity of these pathways, further research is needed to clarify their therapeutic role in many human diseases.

## 6. Fluoroquinolones as Promising Immunomodulatory Agents

Emerging molecular evidence underscores fluoroquinolones as a class of antibiotics exhibiting a pronounced immunomodulatory activity. Specifically, fluoroquinolones have been observed to attenuate the synthesis of pro-inflammatory cytokines such as IL-1β, IL-6, IL-8, and TNF-α by human monocytes in vitro [114]. Furthermore, murine models subjected to LPS exposure have demonstrated that fluoroquinolones induce a reduction in TNF-α serum levels while promoting an increase in IL-10 production [114].

The precise mechanisms through which fluoroquinolones elicit their immunomodulatory effects remain unknown. It has been suggested that these effects might be mediated by the inhibition of phosphodiesterases and the modulation of key transcription factors, including the activator protein 1 (AP-1), the nuclear factor of activated T cells (NF-AT), IL-6, and NF-κB [114,115,116].

For instance, investigations on moxifloxacin administration in mice subjected to cyclophosphamide injection and *Candida albicans* intratracheal inoculation have unveiled its capacity to prevent pneumonia development [117]. This protective effect is associated with marked reductions in TNF-α and IL-8 serum levels compared to control subjects. Complementary immunohistochemical analyses have further revealed a significant attenuation in the expression of NF-κB in the lung tissues of moxifloxacin-treated cohorts. The downregulation of NF-κB could be related to moxifloxacin’s capacity to hinder IκB degradation [117].

Recent investigations have further proposed that ciprofloxacin and levofloxacin could reduce the production of pro-inflammatory cytokines, including IL-1β and TNF-α, by exerting inhibitory effects on the TLR4/NF-κB pathways within microglial cells upon LPS exposure [117]. Correspondingly, moxifloxacin has exhibited the ability to mitigate the rise of IL-6 and TNF-α in LPS-exposed macrophages by downregulating the expression of TLR4 and NF-kB [118].

Complementing preclinical insights, fluoroquinolones have extended their immunomodulatory activity to clinical domains. Notably, norfloxacin is used for secondary prophylaxis against spontaneous bacterial peritonitis (SBP) in cirrhotic patients [119]. In cirrhotic individuals with a history of SBP, orally administered norfloxacin has been associated with the attenuation of NF-κB expression and the reduction in TNF-α, IL12, and IFN-γ serum levels [120]. Importantly, this effect appears unique to norfloxacin, as these changes in cytokine serum levels have not been observed with other antibiotics such as trimethoprim–sulfamethoxazole [120].

Nevertheless, the intricate clinical landscape demands cautious consideration. Gut dysbiosis, the impairment of the intestinal barrier, bacterial translocation, and immune perturbations join together in SBP pathogenesis [121]. Importantly, the dysregulation of the TLR4 signaling pathway achieves a pivotal role in the interplay between gut bacteria and endothelial and immune cells, exacerbating inflammatory pathways and leading to SBP emergence [122].

Preclinical models show that even brief exposure to levofloxacin and norfloxacin alters the gut microbiota, resulting in decreased alpha- and beta-diversity as well as increased pro-inflammatory cytokine production [123,124]. Human data support this trend, notably in the case of ciprofloxacin [125]. Concurrently, fluoroquinolone antibacterial activity limits pathobiont spread, which is intricately linked to dysbiosis, giving anti-inflammatory effects [126]. In fact, fluoroquinolones could have a beneficial role in gut dysbiosis, preventing the expansion of those bacteria which could disrupt the microbiota balance, as demonstrated by a clinical study on small intestine bacterial overgrowth [127]. Furthermore, fluoroquinolones have been shown to inhibit bacterial translocation in cirrhotic patients, which has a preventive effect on SBP and other complications of the disease such as hepatic encephalopathy, variceal bleeding, and hepatorenal syndrome [128,129,130].

## 7. Antibiotics to Modulate Human Microbiota: The Eubiotics

Antibiotics are mainly known to have a disruptive effect on human microbiota. Since most antibiotics have a wide range of effects, they affect both benign and harmful bacteria [131]. Antibiotics’ influence on the gut microbiota is determined by drug-related and host-related factors; among the former, the class, pharmacokinetic and pharmacodynamic features, and, most importantly, the route of administration, dose, and the duration of therapy must be mentioned [131]. An increasing body of research has established a link between the overuse of antibiotics and the emergence of numerous disorders linked to changes in the gut microbiota. Overexposure to antibiotics can also result in the development of genotypic antibiotic resistance in the resident microbiota and its possible transfer to pathogenic bacteria [131]. Several studies assessed gut microbiota alterations following different antibiotic therapies [132]. However, antibiotics can also have a beneficial influence on the microbiota in the gut, a phenomenon known as the “eubiotic” effect. Indeed, it has been shown that some antibiotics promote the growth of helpful microorganisms (Table 1) [10,131].

### 7.1. Rifamixin, a Paradigm for Eubiotics

The most renowned antibiotic for its eubiotic qualities is undoubtedly rifaximin. Rifaximin is a poorly absorbable antibiotic with broad-spectrum coverage (aerobes, anaerobes, and Gram-positive and Gram-negative bacteria), currently used to treat disorders related to changes in the gut microbiota, such as irritable bowel syndrome (IBS), diverticular disease, and hepatic encephalopathy [10]. Several studies have shown that the beneficial effects of rifaximin extend beyond its direct antimicrobial mechanism; these benefits include the ability to modulate the composition of the gut microbiota, alter bacterial virulence, reduce bacterial adherence to the gut mucosa and internalization, and suppress intestinal inflammatory activity through the PXR-NF-κB pathway [133]. Rifaximin does not significantly affect the overall gut microbiota composition, but rather induces an increase in the relative abundance of health-promoting bacteria and modifies bacterial metabolites in the gut [134]. Several studies have been conducted in order to analyze qualitative and quantitative changes in gut microbiota composition after rifaximin treatment. According to a clinical trial based on a multi-tagged pyrosequencing analysis conducted in patients with mild encephalopathy, numerous bacterial networks, including those of *Enterobacteriaceae*, *Bacteroidaceae*, *Porphyromonadaceae*, *Veillonellaceae*, and *Rikenellaceae*, were modulated by rifaximin, and an increase in saturated and unsaturated fatty acids was found in the serum of the treated patients [135]. In a subsequent trial including 15 patients affected by IBS, rifaximin was found to be effective in increasing bacterial diversity, *Firmicutes/Bacteroidetes* ratio, and the abundance of *Faecalibacterium prausnitzii*, a butyrate producer known for its anti-inflammatory properties [136]. Another study conducted on 19 patients with various gastrointestinal and liver diseases demonstrated that treatment with 1200 mg of rifaximin daily for 10 days could boost Lactobacilli abundance without affecting gut microbiota α-diversity [133]. This result may be explained by the fact that certain rifaximin-sensitive bacteria witnessed a minor decrease in abundance that was not statistically significant, while others, such as *Lactobacilli*, gained resistance [10,133]. Beyond its direct antimicrobial mechanism and its role in modulating gut microbiota, rifaximin also has a direct immunomodulatory effect, acting indirectly on NF-kB via the pregnane-X receptor (PXR) and leading to the downregulation of pro-inflammatory cytokines [10,133]. Rifaximin is an intestine-specific human PXR agonist; PXR activation has been proven to attenuate NF-κB signaling, resulting in the weaker expression of pro-inflammatory cytokines (IL-6, IL-10, IL-1β, TNF-α, IFNα). To date, PXR is accounted as a novel target for IBD therapy. Rifaximin, through the abovementioned mechanisms, could represent a tool to rebalance the epithelial barrier and the mucosal immune system, resulting in cell structure and function reconstruction [134]. Another mechanism leading to pro-inflammatory cytokine reduction was demonstrated by Xu et al. in chronic stress rat models [137], in which the oral administration of rifaximin prevented mucosal inflammation by increasing the abundance of *Lactobacilli* and normalizing IL-6 and TNF-α levels. This effect was likely caused by *Lactobacillus* given the reported *Lactobacillus*-induced downregulation of pro-inflammatory cytokines IL-6 and TNF-α in Crohn’s disease [134].

### 7.2. Macrolides: New Perspectives for Lung Microbiota

Recent studies have confirmed that, in addition to its immunomodulatory effects, macrolide therapy can alter the lung microbiome even in the absence of acute respiratory infection [138]. The healthy lung has a very low microbial biomass, and the nature of the lung microbiome differs significantly in quantity and dynamics from those of other body sites with rich microbial communities, such as the gut, skin, mouth, and vagina [139]. Many respiratory disorders previously thought to be microbiologically driven have now been connected to the lung microbiota, although research into causality and probable processes is still in its early stages [139]. Azithromycin influences systemic inflammation, both directly, acting on host immune cells, and indirectly, via the alteration in the lung microbiome [138].

Several studies reported the changes in lung microbiome composition after azithromycin therapy [140,141,142,143]. Antibiotic prophylaxis with azithromycin, as well as ciprofloxacin and doxycycline, in patients with chronic lung diseases was associated with a reduction in alpha-diversity and total bacterial density, with a significantly lower relative abundance of respiratory pathogens such as *P. aeruginosa*, *M. catarrhalis*, and members of the family *Enterobacteriaceae* in sputum [144]. Another clinical trial demonstrated that 250 mg of azithromycin three times weekly for 4 months reduced *Staphylococcus aureus* abundance in the sinonasal microbiota of patients with chronic rhinosinusitis [145]. In 2013, Slater et al. conducted a study [143] on five adult patients with asthma to investigate the longitudinal modifications in the airway microbiome after antibiotic treatment with azithromycin for 6 weeks. Azithromycin was found to decrease the diversity of bacteria in the airways, resulting in changes to the airway microbiome. In some cases, this led to an increased abundance of *Anaerococcus*, a bacterium not typically present in significant numbers. Additionally, the abundance of *Pseudomonas*, *Haemophilus*, and *Staphylococcus*, all known to be associated with airway disease, was reduced. In a subsequent study [142], Segal et al. performed a microbiological analysis on bronchoalveolar lavage fluid in patients with emphysema treated with azithromycin. In this randomized trial, a daily treatment with 250 mg of azithromycin was associated with BALF levels of chemokines and TNF-α, IL-13, and IL-12 reduction, and did not alter lung bacterial burden but reduced alpha-diversity. Interestingly, these changes in the microbiota were associated with an increase in bacterial metabolites, such as indole-3-acetate. Notably, microbiota-derived indole metabolites have an anti-inflammatory role with potential benefits in several inflammatory diseases, such as colitis and ankylosing spondylitis [146,147]. In 2014, a randomized controlled trial [141] was conducted by Rogers et al. with the aim of investigating significant quantitative and qualitative changes in the airway microbiome of patients with non-cystic fibrosis bronchiectasis undergoing long-term, low-dose erythromycin treatment. The study found that erythromycin had no effect on the microbiota composition in patients with a baseline airway infection dominated by *P. aeruginosa.* However, in patients without *P. aeruginosa* airway infection, erythromycin did not appreciably reduce exacerbations and contributed to the displacement of *H. influenzae* by more macrolide-tolerant pathogens, including *P. aeruginosa*. A systematic review and meta-analysis of four randomized controlled trials found that macrolides reduced the levels of *S. pneumoniae*, *Haemophilus influenzae*, and *Moraxella catarrhalis* in children with bronchiolitis. However, no significant reduction in *S. aureus* was observed. Clarithromycin therapy for 3 weeks also decreased serum levels of IL-8, IL-4, and eotaxin. While there was no significant difference in IL-8 serum levels on day 15 between the azithromycin and control groups, a significant reduction in IL-8 nasal lavage levels was reported [140].

Therefore, there are limited data on the effects of macrolides on the lung microbiota. However, given the increasing evidence of their involvement in lung diseases, further research is needed.

### 7.3. Tetracyclines as Potential Eubiotics: Exploring the Evidence

Antibiotics belonging to the tetracycline class have recently attracted much attention because of the correlation between their use and improved blood pressure values in animal models of hypertension. Interestingly, in 2021, a study [148] was carried out to determine the impact of doxycycline antibiotic therapy on blood pressure levels in rat models of hypertension. The rationale for this study was based on evidence of the high prevalence of dysbiosis in hypertension models [149] and on the beneficial effect obtained with antibiotic therapy [150]. Chronic treatment with doxycycline effectively prevented increases in blood pressure, and this result was likely mediated by a doxycycline-dependent decrease in lactate-producing bacteria, as plasma lactate levels are linked to increased blood pressure. Doxycycline also improved colonic integrity by an increase in colonic mRNA levels of the tight junction proteins occludin and zonula occludens 1 (ZO-1), as well as mucin 3 (MUC-3), reducing endotoxemia and LPS serum levels. Evidence supports the pathogenic function of LPSs in hypertension through the activation of the TLR4, leading to the activation of nicotinamide adenine dinucleotide phosphate (NADPH) oxidase and the expression of a series of pro-inflammatory cytokines in the vascular endothelium. Lastly, treatment with the antibiotic doxycycline increased the infiltration of Tregs and IL-10 in the vessel wall. In conclusion, doxycycline regulates the immune response both with and without LPSs by modulating gut microbiota, TLR activity, and cytokine production (i.e., IL-1β and IL-6) [148]. Also, minocycline showed a modulatory activity on the gut microbiota by restoring the *Firmicutes*/*Bacteroidetes* ratio, with a beneficial effect on hypertension [149].

Antibiotics have a fundamental role in modulating the interaction among the several bacterial species inhabiting the human body. Nonetheless, the recent elaboration of the concept of pathobiome, i.e., the complex interaction of pathogens, commensal bacteria and eukaryotes, together with the host, which could be the cause of many diseases, overpassing the idea of one-pathogen–one-disease, could suggest even a novel role for antibiotics [151]. In fact, pathobiome has been associated with immune dysregulation and critical acute diseases (like sepsis) [152]. As shown above, in the OPTIMIZE trial, individuals with cystic fibrosis and *P. aeruginosa* airway colonization treated with azithromycin, together with *P. aeruginosa* eradication, show a significant reduction in their inflammatory state [86]. This trial highlights the role of the pathobiont and immune system relationship in chronic disease, but it lacks consideration about lung microbiota. Indeed, although antibiotic therapy could be effective in several analogous conditions, the interaction between antibiotics and pathobiome is not completely understood and deserves a more in-depth evaluation.

## 8. Clinical Implications of the Complex Interplay Between Pathogens, Host Immune System, and Microbiota

### 8.1. The Clinical Role of Antibiotics Between Chronic Inflammation and Microbiota: Chronic Lung Diseases

Bacteria do not always cause infections in humans. Indeed, trillions of bacteria inhabit human organisms, contributing to maintaining homeostasis; this is the human microbiota. A healthy microbiota is necessary for human metabolism and defense from pathogens [153,154]. On the contrary, each organ of the host can be impacted by the microbiota, and dysbiosis is linked to a wide range of illnesses. In particular, dysbiosis has been implicated in cardiovascular and respiratory diseases, inflammatory bowel diseases, liver disorders, a variety of neoplasms, and metabolic illnesses via the intensification of a chronic inflammatory state [155,156,157]. Indeed, through a complex and mutual crosstalk with the immune system, the microbiota exerts a significant immunomodulatory role dependent on the bacterial balance [158]. Besides their direct immunomodulatory activity, some antibiotics can also regulate the human immune response through their action on the disrupted microbiota. It has been recently proposed that the dysregulation of the lung microbiota could be implicated in chronic respiratory disease exacerbations. Respiratory microbiota composition is influenced by microbial migration, microbial elimination, and the relative reproduction rates of its members. The alteration in one of these factors can induce a condition of dysbiosis. In health, community membership is primarily determined by bacterial migration and elimination, while in chronic lung disease, by regional growth conditions [159]. As a consequence, respiratory dysbiosis could provoke a dysregulated host immune response, which in turn could alter microbial growth conditions in patient airways, further promoting dysbiosis and perpetuating a coupled cycle of inflammation and disordered microbiota [159]. In such a condition, characterized by a bacterial imbalance associated with an abnormal immune response, the role of antibiotics with an immunomodulatory effect could be valuable. To date, few studies have investigated the impact of these antibiotics on the lung microbiota. However, the beneficial effect of low-dose, long-term azithromycin on chronic lung diseases is well known [22]. Even long-term erythromycin therapy has been shown to reduce the exacerbation of chronic obstructive pulmonary diseases [160]. Besides macrolides, doxycycline showed a clinical benefit in chronic pulmonary diseases. Actually, 100 mg doxycycline administered twice daily for 4 weeks improved pulmonary function and reduced inflammatory markers in patients with COPD compared to placebo in a randomized trial [161]. Another randomized placebo-controlled trial showed that the daily administration of 400 mg moxifloxacin for 5 days every 8 weeks, repeated six times, reduced the rate of exacerbations in patients with COPD [162]. Interestingly, none of these patients developed side effects or a significant antimicrobial resistance [162].

### 8.2. The Clinical Role of Antibiotics Between Chronic Inflammation and Microbiota: Gastrointestinal Diseases

These considerations become even more complex in inflammatory human diseases with a strong relationship with gut microbiota alterations. Antibiotics with immunomodulatory action usually have a disruptive effect on intestinal microbiota [163]. However, a combination of ciprofloxacin with a eubiotic such as rifaximin demonstrated a beneficial effect on refractory pouchitis [164]. Further, the pleiotropic effectiveness of rifaximin in inflammatory bowel diseases may be related, beyond its eubiotic effect, to the regulatory properties on intestinal epithelial cells exerted through NF-κB- and pregnane X receptor-related pathways [165]. Nonetheless, when a dysbiotic condition is already set, antibiotic administration could in some conditions contribute to dampening the inflammatory syndrome associated [127]. In general, intestinal inflammation is associated with a reduced gut microbiota diversity, with a noted reduction in *Bacteroides* and *Firmicutes*, thought to have anti-inflammatory properties, and a relative increase in mucosa-associated *Enterobacteriaceae*, including *Escherichia coli* and *Fusobacterium* [166,167]. Rifaximin, restoring the gut microbiota composition, demonstrated a beneficial effect on several intestinal inflammatory conditions such as SIBO [168]. Nevertheless, other antibiotics with immunomodulatory effects and activity against *Enterobacteriaceae* and *Fusobacterium* showed a possible role in this clinical setting. Macrolides and fluoroquinolones alone or in combination therapies showed promising results for the treatment of IBD [169,170]. In particular, the oral administration of azithromycin and metronidazole for 8 weeks effectively resulted in inducing clinical remission in mild–moderate Crohn’s disease [171], while ciprofloxacin alone or associated with biologic therapies or immunosuppressants improved healing in perianal Crohn’s disease [172,173,174,175]. However, a recent clinical trial comparing ciprofloxacin, doxycycline, and hydroxychloroquine therapy to oral budesonide showed no significant differences in inducing Crohn’s disease remission [176]. Nonetheless, individuals treated with ciprofloxacin, doxycycline, and hydroxychloroquine had a longer duration of remission. A recent meta-analysis also reported that adding antibiotics to concurrent medications or standard care therapy enhances the incidence of long-term remission maintenance in ulcerative colitis patients compared to placebo [177].

### 8.3. Antibiotics’ Clinical Function in Chronic Inflammation and Microbiota: Autoimmune Disorders

Autoimmune diseases are intimately associated with immunological dysregulation and changes in the gut microbiome [178]. In fact, *Prevotella copri* and *Lactobacillus* are more prevalent in the early phases of rheumatoid arthritis than *Bacteroidetes*, *Bifidobacteria*, and *Eubacterium rectale* [178]. Conversely, during the active stages of the illness, *Lactobacillus salivarius*, *Collinsella*, and *Akkermansia* become more prevalent, while *Haemophilus* species decrease [178]. PAMPs improve the inflammatory response, the generation of cytokines, and the activation of autoreactive immune cells, even if the precise mechanism linking dysbiosis to rheumatoid arthritis is still unclear. Furthermore, cytokines including TNF-α, IL-6, and IL-1 can cause fibroblasts and other cells to release MMPs and activate NF-κB, both of which contribute to the degeneration of bone and cartilage tissue in rheumatoid arthritis [179].

More complex gut microbiota alterations have also been described to link gut inflammation and spondyloarthritis [180]. Nonetheless, antibiotic therapy has shown effectiveness in rheumatoid arthritis treatment. Tetracyclines, in particular minocycline, were associated with a clinically significant improvement in disease activity, erythrocyte sedimentation rate, and acute-phase reactant serum levels with no absolute increased risk of side effects [181]. Also, levofloxacin was demonstrated to improve rheumatoid arthritis symptoms and to reduce serum inflammatory markers in a clinical trial in which it was evaluated as adjunctive therapy with methotrexate compared to placebo [182]. Furthermore, levofloxacin showed promising results in the treatment of spondyloarthritis. Levofloxacin administration to mice with spondyloarthritis decreased inflammatory infiltrates according to a histological analysis of the ileocolic junction, hind paws, and spines. Moreover, it was associated with a reduction in hyperplasia, goblet cell loss, crypt abscesses, crypt loss, and crypt irregularity in the intestinal mucosa and lower levels of synovial hyperplasia, enthesitis, cartilage neoformation, and ankylosis compared with control mice. A molecular analysis showed that levofloxacin decreased the tissue expression of TNF-α, IL-23a, and JAK3, but also a differential expression of IL-17 between the intestine and the joints. Serum TNF-α was also reduced in levofloxacin-treated mice [183]. The pathogenesis of another autoimmune disease, such as multiple sclerosis, has been associated with gut microbiota dysregulation. In particular, the abundance of certain bacteria, such as *Flavonifractor plautii*, which is increased in active disease, correlates with inflammatory markers like blood leukocyte levels, CRP, and IL-17A and IL-6 overexpression in white blood cells. Bacterial species that were more abundant in disease-active treatment-naïve multiple sclerosis were positively linked to a group of plasma cytokines including IL-22, IL-17A, IFN-β, IL-33, and TNF-α [184]. Moreover, MMPs seem to be implicated in multiple sclerosis development; in particular, MMP-2, -8, and -9 could enhance the demyelination process favoring neutrophils and T cell migration through the blood–brain barrier [185]. Individuals with multiple sclerosis treated with oral minocycline observed a considerable decrease in the number of gadolinium-enhancing magnetic resonance imaging lesions as well as a drop in MMP-9 blood levels [186,187]. Additionally, minocycline lowers the relapse rate in people with relapsing–remitting multiple sclerosis [183]. Furthermore, when combined with glatiramer acetate, minocycline reduces neurological abnormalities, shown by magnetic resonance imaging [188]. Based on these findings, Metz et al. treated 72 individuals who had experienced a first demyelinating episode with 100 mg of oral minocycline twice a day. Compared to the placebo group, minocycline-treated individuals had a markedly decreased risk of multiple sclerosis after six months [189]. Although less investigated, even doxycycline, when administered in association with IFN beta-1a, reduces contrast-enhancing lesion numbers [186]. Interestingly, these beneficial effects seem to correlate with the decrease in serum MMP-9 levels [186].

### 8.4. Side Effects and Antimicrobial Resistance: Limitations of a Long-Term Antibiotic Therapy

Although several antibiotics have demonstrated potential in the management of chronic inflammatory conditions, the need for long-term antibiotic therapy for chronic illnesses raises questions regarding possible adverse effects. In this regard, a lot of research has been carried out, especially with azithromycin. There may be a link between long-term azithromycin medication and hearing damage according to a meta-analysis of six randomized controlled studies [190]. It is interesting to note that as compared to a placebo, using azithromycin did not increase the frequency of gastrointestinal side effects. The long-term use of antibiotics may also raise the risk of antibiotic resistance [190]. A recent clinical trial evaluating the long-term effect of low-dose azithromycin therapy in individuals with HIV and chronic lung diseases for 48 weeks showed an increase in *S. pneumoniae* and *S. aureus* strains resistant to azithromycin and tetracyclines in nasopharyngeal swabs and sputum compared to placebo [191]. However, resistant *S. pneumoniae* was not found 24 weeks after treatment withdrawal, while an *S. aureus* resistant strain persisted [191]. Side effects of long-term tetracyclines therapy have been evaluated in the treatment of acne. In particular, minocycline seems to be associated with an increased risk of liver dysfunction and systemic erythematous lupus development compared to placebo [192]. These adverse events seem to not be associated with doxycycline or other tetracyclines during long-term therapy [193]. Nonetheless, low-dose, long-term doxycycline therapy is generally well tolerated as malaria chemoprophylaxis [193]. Conflicting data exist about the impact of long-term tetracycline use on the development of antimicrobial resistance [194,195,196,197]. Actually, a systematic review showed that 2–18 weeks of doxycycline, tetracycline, oxytetracycline, or minocycline therapy was associated with an increase in tetracycline resistance b, including in oral, intestinal, and upper respiratory tract microbiota. On the contrary, 18 weeks of oxytetracycline/minocycline therapy did not increase tetracycline-resistant *Propionibacterium* abundance in skin microbiota [198]. Long-term tetracycline therapy could also induce non-tetracycline resistance. However, no significant increase in commensal bacteria in the gut microbiota was detected in three studies. Nonetheless, the impact on AMR of tetracycline’s long-term use in sexually transmitted disease prophylaxis and malaria prophylaxis remains unclear since the available studies do not properly consider the prevalence of tetracycline resistance before long-term treatment [198]. Fluoroquinolone use has been linked to several adverse effects. Tendon rupture; arthralgia; tendonitis; gait disturbance; peripheral neuropathies; fatigue; memory loss; depression; sleep disorders; diminished taste, smell, vision, and hearing; phototoxicity; genotoxicity; QTc prolongation; hematological effect; hepatic eosinophilia effect; pulmonary interstitial eosinophilia; immunological side effects; hypoglycemia; and CYP 450 inhibition are among the most severe of them [199]. Due to these disabling and potentially permanent adverse effects, the FDA and EMA have recently limited the recommended use of fluoroquinolones [199]. Furthermore, there is no doubt that their widespread use contributes to a selection pressure that promotes antimicrobial resistance [200,201,202]. Nonetheless, the reduction in fluoroquinolone use, following the FDA and EMA recommendations, has not been associated with a decrease in fluoroquinolone resistance levels [203]. Indeed, a retrospective analysis of neutropenic patients with hematological malignancies who received long-term levofloxacin prophylaxis did not show an increase in Gram-negative bacteria resistant to fluoroquinolones during the treatment. On the contrary, the rate of Gram-negative bacteria resistant to fluoroquinolones producing extended-spectrum-beta-lactamase colonization significantly increased after prophylaxis was discontinued [204]. Rifaximin, on the contrary, seems to be safe and tolerable for long-term treatment, as demonstrated by a 24-month randomized clinical trial with a placebo [205]. Moreover, although existing evidence of rifaximin resistance development raises a concern [206], the risk of developing resistance seems to be lower than other absorbable antibiotics and less transmittable, being chromosome-mediated and not plasmid-mediated [133].

As a matter of fact, pleiotropic antibiotics disclose a great therapeutic potential in clinical practice. Nonetheless, side effects and antimicrobial resistance development could limit their use and require both deeper knowledge and a careful balance between the risks and benefits of their administration.

## 9. Conclusions

Numerous preclinical and clinical studies have shed light on the immunomodulatory effects of macrolides, tetracyclines, and fluoroquinolones, which exert their influence on both innate and adaptive immune responses. These effects may arise from the intricate interplay between these antibiotics and the human microbiota. While the complexity of this relationship necessitates further research to uncover its various aspects and molecular mechanisms, some of these drugs have already secured approval for their immunomodulatory properties.

The impact of antibiotics on the human microbiota continues to be a subject of ongoing investigation. Rifaximin stands out as a recognized eubiotic agent, while other antibiotics, particularly macrolides and tetracyclines, are more closely associated with the onset of gut dysbiosis. Recent findings concerning the lung microbiota and its relevance to human diseases underscore the importance of addressing the role of antibiotics in this context, where both acute and chronic conditions may benefit from their properties.

These antibiotics find application across a wide spectrum of clinical settings, encompassing acute infectious syndromes, chronic lung diseases, and systemic autoimmune disorders. However, concerns about potential side effects and the development of antimicrobial resistance emphasize the need for their cautious utilization and the selective targeting of patients who stand to derive the greatest benefits. Among these antibiotics, macrolides, tetracyclines, and rifaximin offer greater safety for long-term use, whereas the extended administration of fluoroquinolones presents more significant challenges, especially regarding microbial resistance and adverse events. Nonetheless, the pleiotropic effects of antibiotics present a captivating area of study, casting them as mediators in the intricate relationship between microbiota and human cells. The complexity of these molecular and cellular interactions, coupled with their clinical implications, constitutes a significant scientific challenge with immense research potential.

## Figures and Tables

**Figure 1 antibiotics-13-01176-f001:**
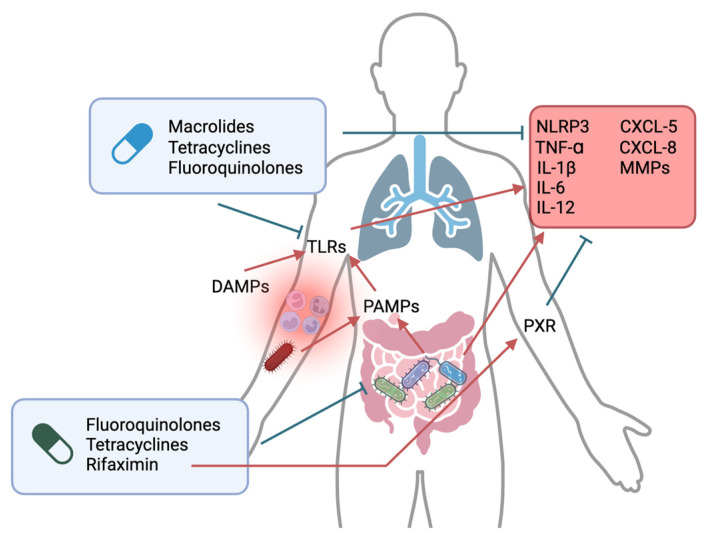
Overview of immunomodulatory effect of antibiotics at a molecular level. Macrolides, tetracyclines, and fluoroquinolones reduce Toll-like receptor (TLR) expression by immune cells and lead to the inhibition of pro-inflammatory cytokine production, chemokines, and matrix metalloproteinases (MMPs), which play a significant role in enhancing both acute and chronic inflammation. Furthermore, some antibiotics, such as rifaximin, tetracyclines, and fluoroquinolones, indirectly modulate immune response by modulating the gut microbiota. Nonetheless, rifaximin demonstrates a specific anti-inflammatory effect by agonizing the pregnane X receptor (PXR). CXCL: chemokine (C-X-C motif) ligand; DAMPs: damage-associated molecular patterns; IL: interleukin; NLRP: NOD-like receptor protein 3; PAMPs: pathogen-associated molecular patterns; TNF- α: tumor necrosis factor α.

**Table 1 antibiotics-13-01176-t001:** The “eubiotic” effect of antibiotics. Beneficial influence of different antibiotic classes in quantitative and qualitative features of microbiota.

*Antibiotic*	Quantitative Changes	Qualitative Changes
*Rifaximin*	Gut microbiome: ↑ α-diversity	Gut microbiome: ↑ *Firmicutes*/*Bacteroidetes* ratio ↑ *F. prausnitzii* ↑ *Lactobacilli abundance*
*Macrolides*	Airway microbiome: ↓ α-diversity and richness	Airway microbiome: ↓ Respiratory pathogens (*P. aeruginosa*, *M. catarrhalis*, *H. influenzae*, *S. aureus*, *S. pneumoniae*)
*Tetracyclines*		Gut microbiota: ↓ lactate-producing bacteria Restoration of *Firmicutes*/*Bacteroidetes* ratio

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
