# Peer review of "“Pleiotropic” Effects of Antibiotics: New Modulators in Human Diseases"

_antibiotics, 2024, doi:10.3390/antibiotics13121176_

Round 1
Reviewer 1 Report
Comments and Suggestions for Authors
The multiple effects due to use or prolong use of antibiotics offer new therapeutic opportunities by interacting with human cells, signaling molecules, and bacteria involved in non-infectious diseases like spondylarthritis and inflammatory bowel diseases. In this connection, the authors have undertaken a legitimate problem but the study lacks pertinent informations.
The queries are as follows:
- In the title and abstract the word ‘pleiotropic’ may be replaced. It is used mainly for the genetic phenomenon where a single gene affects multiple traits or phenotypes that appear to be unrelated as the review focuses on multiple effect of antibiotics on the human body not the genetic regulation.
- It will be worthy to mention the purpose behind this review. The inclusion and exclusion criteria of various antibiotics may also be elaborated.
- Creation of different review sections by including few studies conducted by the authors or available elsewhere may make the review more informative.
- The authors have focused only on the microbiome of the various human patients. The elaboration of pathobiome role may help in better understanding of review.
Abstract Section
- Line 22: The line “Rifaximin, for example, helps maintain the balance between immune cells, epithelial cells, and gut bacteria”; may be reframed for better readability.
Introduction Section
- Line 37: The line “Antibiotic consumption has increased over time, also due to factors such as rising life expectancy and the complexity of comorbidities”; may be reframed for better readability.
- Line 43: The line “Some antibiotics, however, carry auxiliary features contributing to their extensive clinical usage these additional positive benefits, known as pleiotropic effects, may be related or independent to the primary mechanism of action of the drug and are frequently unexpected or unplanned”; may be reframed for better readability.
Side effects and antimicrobial resistance: limitations of a long-term antibiotic-therapy
- This section may be elaborated as the authors have not included role of the long-term antibiotic therapy on antimicrobial resistance. It may be elaborated.
Author Response
Thank you for the comment, it is very constructive and it greatly enriches the quality of our review.
- We have modified the title to enhance clarity. We understand the reviewer’s concerns regarding the term “pleiotropic”. Nonetheless, we believe that it remains appropriate given its Greek roots consisting of "pleion" meaning "very many" and "tropy" meaning "manner". In addition, the term has been adapted in pharmacology since Davignon et al. used it to describe the diverse effects of statins. Therefore, we feel that “pleiotropic effects” could convey the multiple impacts of antibiotics in this context.
- We have clarified the purpose of the review. Additionally, we have included the antibiotics’ choice criteria, explaining our focus on antibiotics known to have non-antimicrobial effects.
- We have implemented some review sections with relevant studies to improve the review’s informativity.
- We have expanded our analysis to include the role of the pathobiome, especially in the context of inflammatory diseases. This addition offers a more comprehensive view of how antibiotics interact with both the microbiome and pathogenic organisms, contributing to disease progression and therapy.
- We have changed the sentence of line 22 to improve readability.
- We have changed the sentence of line 37 to improve readability.
- We have changed the sentence of line 43 to improve readability.
- We have expanded this section to address the role of long-term antibiotic therapy in promoting antimicrobial resistance.
Reviewer 2 Report
Comments and Suggestions for Authors
Dear Editor,
I recommend to accept the manuscript antibiotics-3302756 in its present form.
The main question addressed by this review article is to explore the pleiotropic potential of antibiotics, from molecular and cellular evidence to their clinical application, in order to optimize their use.
This review article emphasizes that the understanding of these effects is essential to ensure careful use, particularly in consideration of the threat of antimicrobial resistance.
This review article adds to the subject area compared with other published material the comprehensive overview of very important topics described in details in chapters such as “Immunomodulatory effect of antibiotics in critical illnesses”, “Antibiotic modulation of chronic inflammation” with very illustrative figure 1, followed by overview of macrolides, tetracyclines and fluoroquinolones impact on immunomodulation. The important quality of this manuscript is overview of antibiotics with eubiotics properties. Chapter describing the complex interplay between pathogens, host immune system, and microbiota is very well written and important for continual education of medical specialists.
The topic is original and relevant in the field.
The type of article is well selected.
The title is informative and it relates to the content of the article.
Keywords are appropriate and reflect the content of the article.
The Abstract is not structured, since this manuscript is the review article.
Scientific content of this article is relevant and well presented.
The Conclusions are consistent with the evidence and arguments presented and they address the main question posed.
The references are appropriate, they are up-to-date and numbered consecutively in the manuscript.
Author Response
Thank you for your comment. We greatly appreciate your analytic feedback.
Reviewer 3 Report
Comments and Suggestions for Authors
The proposed manuscript examines in detail the long-known pleiotropic effects of a number of antibacterial agents. A systematic review with evidence provides excellent insight into these properties. The manuscript will be a useful aid to practicing clinicians in choosing a specific antibiotic preparation for their patients. The manuscript is fully within the scope of the journal and I offer acceptance for publication.
Author Response

(The authors gave the same response as above.)
